# Combining Cognitive and Affective Measures with Epistemic Planning for Explanation Generation

**Ronald P. A. Petrick**
Department of Computer Science
Heriot-Watt University
Edinburgh, Scotland, United Kingdom
R.Petrick@hw.ac.uk

**Sara Dalzel-Job** and **Robin L. Hill**
School of Informatics
University of Edinburgh
Edinburgh, Scotland, United Kingdom
sdalzel@exseed.ed.ac.uk,R.L.Hill@ed.ac.uk

## Abstract

This paper presents an overview of the EPSRC-funded project Start Making Sense, which is investigating explainability and trust maintenance in interactive and autonomous systems. This project brings together experimental research in cognitive science involving cooperative joint action with the practical construction of automated planning tools to apply to the task of explanation generation. The project's challenges are addressed through three concrete objectives: (i) to study cooperative joint action in humans to identify the emotional, affective, or cognitive factors that are essential for successful human communication, (ii) to enhance epistemic planning techniques with measures derived from the studies for improved human-like explanation generation, and (iii) to deploy and evaluate the resulting system with human participants. We also describe initial work from the cognitive side of the project aimed at exploring how ambiguity, uncertainty, and certain types of biometric measurements impact instruction giving and explanation actions in scenarios with humans. The insights from this work will be combined with epistemic planning techniques to generate appropriate explanatory actions in similar instruction giving scenarios.

## Introduction

A fundamental problem in the design of autonomous systems is that of action selection: based on the current state of the world, what action should the system take in order to achieve its goals? In the presence of humans, this problem typically becomes more complex: the system may also need to reason about the states, actions, and intentions of these agents. In collaborative environments that involve human communication, it is particularly important to identify, interpret, and understand the multimodal affective signals that humans employ, and which are often necessary for effective, successful achievement of communicative goals.

For instance, consider a tourist on a guided walking tour of a city. After reaching a place where they can see they are almost back to the starting point, the tour guide says "Let's go up that hill," pointing to a large hill. "We can get a good view of the city from there." However, on seeing the tired expression on the tourist's face, the guide adds "Or we can stop at that cafe over there and take a break." This scenario has two important features. First, it demonstrates that people like to be aware of their context and know what is going on. This is especially true in situations where a decision may not

have been anticipated or expected. Here, an explanation may be needed not only to justify a decision but also to establish confidence in that choice: in other words, to trust it. Second, being able to read the situation and adapt to the needs of the moment is important when considering the possible actions that could be taken in a given situation. Here, a decision may need to be made dynamically. These two features capture the idea of dynamic trust maintenance, which will be needed for a broad range of the AI systems that are expected to be deployed in the near future, e.g., automated vehicles, service robots, or interactive voice-based assistants.

This paper presents an overview of the EPSRC-funded project Start Making Sense: Cognitive and Affective Confidence Measures for Explanation Generation Using Epistemic Planning,[1] which is investigating the need for explainability and trust maintenance in interactive and autonomous systems. To do so, this project brings together experimental research in cognitive science involving cooperative joint action with the practical construction of automated planning tools, in particular epistemic planning techniques, to apply to the task of explanation generation. This challenge is being addressed by tackling three key objectives: (i) to study cooperative joint action in humans to identify the emotional, affective, or cognitive factors that are essential for successful human communicative goals; (ii) to enhance epistemic planning techniques with measures derived from the cognitive science studies; and (iii) to deploy and evaluate the effectiveness of the resulting system with human participants in situations that require explanation.

Central to this work is the idea of understanding the affective measures that humans use during activities like instruction giving, plan following, and explanation generation; both when communication is successful but also when it fails. The goal is to characterise these measures in a form that enables them to be combined with tools based on epistemic planning, an approach that models the changing beliefs of the planner and other agents during the plan generation process. Affective measures will therefore help guide the planner's generation process, for instance as a special type of domain control knowledge or heuristic state information, enabling the planner to use this information not only for task-based action selection, but also to plan appropriate

---

[1]http://start-making-sense.org/

actions for communicative goals such as explanation generation, possibly as a result of dynamic changes in the interactive context. As a result, this work is also situated in the area of explainable planning, a subarea of the recent trend of research in explainable AI.

In the remainder of the paper we outline our approach and the project's main objectives, directions, and goals.

## Related Work

Recent developments in artificial intelligence and machine learning research, such as deep learning, are seen to have resulted in dramatic improvements in prediction and accuracy, but often at the expense of human interpretability. As a result, there has been a rapid growth in research under the general banner of explainable artificial intelligence (XAI), typified by grant funding schemes like DARPA's XAI programme (DARPA 2016) or EPSRC's Human-Like Computing Strategy Roadmap (EPSRC 2017), which seek to address a core need in future systems and to respond to challenges like the EU's "Right to Explanation" initiative (Goodman and Flaxman 2017). Understandably, much work has focused on the specific concerns around, e.g., deep learning, rather than on the general need for human-like explanation.

This project instead builds on research in explainable planning (XAIP) (Fox, Long, and Magazzeni 2017), a subarea of XAI and automated planning (Ghallab, Nau, and Traverso 2004). While techniques like machine learning make decisions based on mined data, automated planners traditionally build plans of action by using symbolic causal models combined with search techniques. XAIP seeks to address the challenges of XAI—to build trust and transparency when interacting with humans (see, e.g., (Miller 2017))—by leveraging automated planning models and recognising the role that humans play in the planning loop with respect to systems deployed with such tools.

XAIP is fast becoming an established recent direction in the planning community, with a first workshop dedicated to this topic (Magazzeni et al. 2018) appearing at the 2018 International Conference on Automated Planning and Scheduling (ICAPS).[2] While some of the underlying ideas concerning planning and explainability have a longer history (Sohrabi, Baier, and McIlraith 2011; Seegebarth et al. 2012), more recent approaches like (Chakraborti et al. 2017; Sreedharan, Chakraborti, and Kambhampati 2017), have resulted in new directions and new planning algorithms.

## Approach

In contrast to most approaches in XAIP, our work combines research from cognitive science on affective measures used in human communication, together with recent techniques in automated planning, notably epistemic planning. In this section, we briefly highlight the key ideas from these two areas and how they are being brought together.

### Affective Measures in Human Communication

From the cognitive perspective, we build on the view that effective explanations arise from cooperative joint action and efficiently satisfy the goals of human communication. For an agent (human or artificial) involved in an interaction, information exchanged needs to lead to acceptance, comfort, and trust in their communication partner, if they are to successfully influence the interlocutor's actions. However, the establishment of trust also depends on the shortcuts, heuristics, and spontaneous choices that people make in interactions, which are often based on emotional or affective factors. As in affective computing more generally, an artificial agent needs to: (i) detect the signals of its interlocutor's affective state; (ii) interpret and understand the meaning behind those signals to infer conclusions; and (iii) be able to take appropriate actions which measurably influence that state.

While confidence and comfort can lead to trust in technologies (Nass and Brave 2005), there is evidence from instruction giving experiments like the HCRC Map Task (Anderson et al. 1991)[3] that have shown that speakers typically under-explain until things go wrong. Paradoxically, successful response to such a failure may build greater confidence in listeners than a consistently verbose explanatory approach which minimises failure rate; similarly, some explanation or correction strategies can be disorienting or annoying (Foster et al. 2009; Henderson, Matheson, and Oberlander 2012). Thus, an important aspect of this project is to understand and develop the capacity to diagnose and repair failures which may be signalled only through brief facial expressions, head movements, or altered body posture.

To this end, the project is conducting user studies with human participants to understand the affective measures that are helpful for effective explanation. Initially, data from previous projects is being used, to study preferred styles in human explanation generation (Carletta et al. 2010) and error detection (Hill and Keller 2014). We are also analysing results on facial expressions and eye-tracking from a current project attempting to measure believability in political messages and what motivates people to agree with, and disseminate, them (Cram 2017). The goal is to synthesise these approaches, adopting mixed methods to collect objective (biometric) and subjective (probe questions) data from individual participants in new experiments. By capturing and annotating conventional linguistic dialogue with paralinguistic signals (e.g., intonation, hesitation, gesture) and behavioural signals (e.g., facial expressions), we expect to identify the best predictors of moment-by-moment shifts in levels of acceptance, comfort and trust that then be used to help build more intelligent artificial systems.

### Initial Phase of the Empirical Cognitive Research

The first phase of empirical cognitive research on the project aims to confirm the range of human social signals, affective responses, and behavioural patterns exhibited during co-operative joint action in a shared audio-visual environment. As a starting point, an instruction giving and following scenario is considered based on an enhanced version of the HCRC Map Task (Anderson et al. 1991) paradigm, to simulate navigation with instruction giving and following. In the standard version of the Map Task, an instruction giver

---

[2]http://icaps18.icaps-conference.org/xaip/

[3]http://groups.inf.ed.ac.uk/maptask/

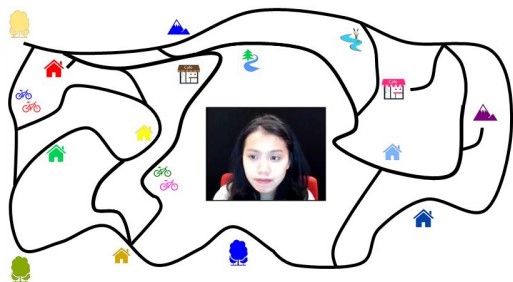

Figure 1: A sample instruction giver map from the enhanced Map Task with an image of a human follower displayed.

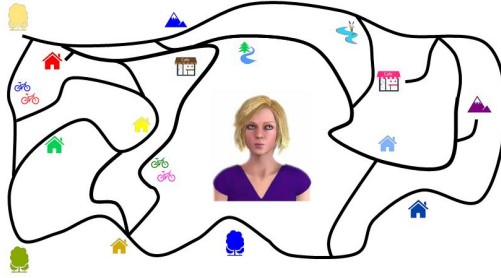

Figure 2: A sample instruction giver map from the enhanced Map Task with an artificial avatar follower displayed.

guides an instruction follower around a map using landmarks. Importantly, the maps of the instruction giver and follower are not aligned: landmarks may be in different locations or in some cases completely different landmarks are present. The task therefore provides opportunities to study how humans communicate and recover from problems in such scenarios. In our enhanced version, we make one critical change: the interlocutors will be clearly visible to each other. These experiments will be used to determine the indicators of successful (or unsuccessful) communication produced during the dynamic process rather simply generating a measure upon completion of the task. In other words, we are proposing a system of continuous monitoring and measurement suitable for agile prediction and adaptive planning.

In the first round of experiments, human participants will observe another human comprehending and responding to their directions in real-time (see Figure 1). Thus, the instruction giver can modify the dialogue based on behavioural feedback and determine the level of success for themselves, e.g. whether the instruction follower is looking at the correct target, has a confused expression or exhibits some other cue. The next set will examine whether human-like behaviour is sufficient to produce the same results or whether every aspect of the interaction needs to be human. To begin understanding this side of the interaction process the same basic paradigm will be adopted, only this time the instruction follower will appear as an artificial avatar (see Figure 2). Critically, however, the avatar is actually generated from the responses of a human recorded by a video camera or webcam. Thus, while the responses may appear to be artificial,

```
action ask-location(?a : agent)
   preconds: K(interact = ?a) &
             !K(requestLoc(?a)) &
             !K(otherAttentionRequests)
   effects:  add(Kf,requestLoc(?a)),
             add(Kv,request(?a))

action give-directions(?a : agent, ?l : loc)
   preconds: K(interact = ?a) &
             K(requestLoc(?a)) &
             Kv(request(?a)) &
             K(request(?a) = ?l) &
             !K(otherAttentionRequests)
   effects:  add(Kf,requestAnswered(?a))
```

Figure 3: Actions for a direction giving agent.

the underlying behaviour is genuinely human and presented through what is essentially a motion-capture system. Ultimately, synthetic stimuli which are generated from our computational models and contingent on the evolving interaction and dialogue will also be tested. These artificial agents will need to portray authentic communication by combining both the fundamental aspects of two-way interaction (comprehension and production): correctly interpreting and understanding observed human behaviour; as well as displaying appropriate human-like reactions.

## Epistemic Planning

The main technical tool employed in this project is a recent approach to automated planning (Ghallab, Nau, and Traverso 2004), called epistemic planning (Bolander 2017), which can be used for action selection in state-based, goal-directed systems that operate in the presence of other agents (human or artificial). Traditional automated planners focus on solving the problem of finding an ordered sequence of actions (a *plan*) that, when chained together, transform an initial state into a state where a set of specified goal objectives are achieved. Planning problems are usually described in a symbolic form that specifies the objects, actions, states, and goals that make up the planner's operating environment. A central goal of planning research is to build general purpose or domain-independent planning systems that are able to solve a range of planning problems in different domains, rather than just a single problem in a particular domain.

Epistemic planning builds on standard automated planning approaches and attempts to model how the knowledge and beliefs of agents evolve during the planning process. In this project, plans are generated using PKS (Planning with Knowledge and Sensing) (Petrick and Bacchus 2002; 2004), an early epistemic planning system.

For instance, Figure 3 shows an example of two actions defined in PKS's modelling language for a simple direction giving agent. Here, `ask-location` models an information-gathering action that asks another agent for a location they are trying to reach, while `give-directions` describes an action for supplying a response when such information is provided. Actions are described by their preconditions (the conditions that must be true for an action to be applied) and

| Plan | Description |
|------|-------------|
| `greet(a1)` | Greet agent `a1` |
| `ask-location(a1)` | Ask `a1` for a location |
| `ack-request(a1)` | Acknowledge `a1`'s request |
| `give-directions(a1,`
`    request(a1))` | Respond to `a1`'s request |
| `bye(a1).` | End the interaction |

Table 1: A plan for giving directions to an agent.

```
action ask(?x,?y,?p)
    preconds: ¬K[?x]?p & K[?x]K[?y]?p
    effects:  add(Kf,K[?y]¬K[?x]?p)

action tell(?x,?y,?p)
    preconds: K[?x]?p & K[?x]¬K[?y]?p
    effects:  add(Kf,K[?y]?p)
```

Figure 4: Actions with nested multiagent beliefs.

their effects (the changes the action makes), where references like `K(...)` are queries of the planner's beliefs. The planner uses these actions to form plans by chaining together ground actions instances to achieve the goals of the planning problem. Table 1 shows a possible plan that could be generated by PKS for interacting with a human agent requesting directions to a given location.

An important feature of most epistemic planners is their ability to reason with multiagent beliefs: information about other agents that is often nested (e.g., "agent A believes agent B believes P") (Fagin et al. 1995). This is a challenging problem for automated planners which must provide a solution that is both expressive enough to model a variety of problems while being efficient enough to be implemented in a manner that does not negatively affect the plan generation process. PKS does this by restricting the form of the representation used by the planner and keeping the reasoning language simple (Steedman and Petrick 2007). An example of actions encoded in this way is given in Figure 4 (where `K[?x]p` denotes the idea that "agent `?x` knows `p`").

Epistemic planners like PKS therefore provide us with powerful tools for building systems that can perform action selection with complex reasoning about other agents and their beliefs. For instance, PKS has previously been used for generating plans in task-based scenarios that require socially-appropriate human-robot interaction (Petrick and Foster 2013), and that involve multiple humans.

### Explanation Generation

The main technical contribution on this project is to use and enhance an epistemic planner like PKS with intuitions from the cognitive science studies to generate plans which inherently contain more human-like explanations. The goal is to not only generate interactive plans like in Table 1, but to also generate the necessary plan explanations, if required, during the epistemic planning process. Since epistemic planners are capable of reasoning about the beliefs of other agents, we can use the planner's belief about a human agent's knowl-

edge (or lack thereof) of different steps in a plan to automatically drive the explanation generation process. (For instance, if the `give-directions` action involves locations that are believed to be unknown to the human, appropriate explanation can be built into the plan.)

The key technical challenge here is to make the generation process fast while still producing high-quality plans. To do this, the identified affective measures from the cognitive science studies will serve as a type of heuristic to inform and guide the plan generation process and appropriately rank the generated plans. Part of this work therefore involves identifying an effective set of measures that can be captured in the planner's representational language, to ensure that relevant scenarios can be modelled with the planner.

Evaluation of the resulting system with human participants is also essential to establish that the resulting plans achieve their expected behaviour. While we will perform standard planning benchmark tests to establish the correctness, quality, and efficiency of the resulting planning system, we also aim is to keep the human in the loop throughout, using situations like the tour guide or direction giving scenarios, to evaluate response detection and adaptive planning techniques using Wizard-of-Oz and Ghost-in-the-Machine experiments (Janarthanam et al. 2014; Loth et al. 2015).

### Conclusions

Combining planning and human interaction, especially in collaborative settings, presents several important challenges that must be addressed due to the necessary presence of humans in the planning loop (Kambhampati and Talamadupula 2015): human activities must be taken into consideration, plans must ensure humans and artificial systems are able to work together effectively for efficient task completion, and plan decisions and effects should be communicated in a manner that improves trust and transparency. This project aims to make contributions in all of these areas. Most importantly, this project places understanding the human experience at the heart of its approach to building tools for explainable epistemic planning, and we have taken a first step in this direction through a series of initial experiments based on the HCRC Map Task paradigm. By basing our technical extensions on affective insights from human studies, and evaluating the result of our tools on human participants, we believe the resulting epistemic planning tools will lead to interactive and autonomous systems that are better prepared for and more acceptable to the expectations of humans.

### Acknowledgements

We are indebted to our friend and colleague, Jon Oberlander (`http://homepages.inf.ed.ac.uk/jon/`), who helped shape this project but whose unexpected passing in 2017 means that we must proceed without his invaluable collaboration. Jon will be sorely missed and this project is dedicated to his memory.

This work is funded by the UK's EPSRC Human-Like Computing programme under grant number EP/R031045/1.

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
