# OpenReview forum: "Combining Cognitive and Affective Measures with Epistemic Planning for Explanation Generation"
_icaps-conference.org/ICAPS/2019/Workshop/XAIP — XAIP 2019_

### Official Review · AnonReviewer2 · 2019-04-25
**Project proposal on cognitive science & epistemic planning in instruction giving; is this XAI?**

**Rating:** 3
**Confidence:** 2

**Review:**

The paper essentially is an EPSRC project proposal, somewhat reshaped (so far as I can tell) into a workshop submission.

This strikes me as unusual (this is the first time I see a project proposal turned into a paper), and there are no concrete results to discuss. However the ideas and approach could be useful & interesting for discussion at the workshop so I'm fine with that.

Proposal summary in my words: address instruction-giving problems involving the interpretation of, and dynamic reacton to, human signals such as facial expressions. Conduct cognitive sciene experiments to determine the relevant human aspects and metrics; use epistemic planning to generate instructions taking other agents' knowledge into account.

This strikes me as an interesting instruction-giving project. What's not qiute so clear to me is, what is the link to XAI?

One can call an "instruction" an "explanation", and the paper/proposal frequently does so. But does that mean that instruction giving is explainable AI? My understanding of XAI is that what we're trying to explain are the decisions taken by an AI system. In instruction giving (in general, and in this paper in particular), instead an AI system explains some third artifact X (external to the AI system) to a human. This seems to be a quite different problem. The strongest connection I can see is that both ultimately require communication with human users, incurring (potentially?) similar cognitive considerations on that side.

In short, the paper proposes to use planning techniques for generating explanations (i.e., instructions) of something outside the planner; as opposed to explaining the decisions of the planner itself. I hence find the link to XAIP, and hence the workshop, a bit tenuous.

It could be interesting to discuss these things at the workshop though, so I'm not adverse to including the paper. If included, I'd appreciate if the authors could discuss the above points, in the paper and their presentation.

Small point: In the motivating example in the introduction, it did not become clear to me what the role/importance is of "After reaching a place where they can see they are almost back to the starting point" (one could take a break in a cafe at any point on a tour).

---

### Official Review · AnonReviewer1 · 2019-05-09
**Interesting Position Paper discussing combining Epistemic Planning and HSR to pursue effective explanations**

**Rating:** 4
**Confidence:** 3

**Review:**

I'm not a fan of seeing the abstract almost verbatim later on in the introduction section. I get it, but why have it in here twice?

Related Work:
  I'm surprised you don't have any mention of Epistemic Planning here yet, or how it ties in. What part of Epistemic Planning is this paper going to take? Can I just use it out-of-the-box for Explanation Generation?

Epistemic Planning:
 There's been some progress since 2004 in the realm of epistemic planners and their limitations. Notably a couple AAAI papers around 2010/2012 that discuss depth limitations to nested knowledge. Merely citing  Bolander and the Fagin Book (from 1995) is a bit, dismissive of the current field of Epistemic Planning. Were this a full paper, I'd want a more up-to-date discussion.


Minor:
  You really like footnotes.

---

### Decision · Program_Chairs · 2019-05-15

**Decision:**

Accept

**Comment:**

The reviewers agree to accept. Please address all review criticism as best possible for the final paper version and its presentation at the workshop. Looking forward to discuss your work at the workshop!